# Drug-Resistant Fungi: An Emerging Challenge Threatening Our Limited Antifungal Armamentarium

**DOI:** 10.3390/antibiotics9120877

**Published:** 2020-12-08

**Authors:** Amir Arastehfar, Toni Gabaldón, Rocio Garcia-Rubio, Jeffrey D. Jenks, Martin Hoenigl, Helmut J. F. Salzer, Macit Ilkit, Cornelia Lass-Flörl, David S. Perlin

**Affiliations:** 1Center for Discovery and Innovation, Hackensack Meridian Health, Nutley, NJ 07110, USA; amir.arastehfar@hmh-cdi.org (A.A.); Rocio.GarciaRubio@hmh-cdi.org (R.G.-R.); 2Life Sciences Programme, Supercomputing Center (BSC-CNS), Jordi Girona, 08034 Barcelona, Spain; toni.gabaldon@bsc.es; 3Mechanisms of Disease Programme, Institute for Research in Biomedicine (IRB), 08024 Barcelona, Spain; 4Catalan Institution for Research and Advanced Studies. Pg. Lluís Companys 23, 08010 Barcelona, Spain; 5Department of Medicine, University of California San Diego, San Diego, CA 92103, USA; jjenks@health.ucsd.edu; 6Clinical and Translational Fungal-Working Group, University of California San Diego, La Jolla, CA 92093, USA; mhoenigl@ucsd.edu; 7Division of Infectious Diseases and Global Public Health, Department of Medicine, University of California San Diego, La Jolla, CA 92093, USA; 8Section of Infectious Diseases and Tropical Medicine, Department of Internal Medicine, Medical University of Graz, 8036 Graz, Austria; 9Department of Pulmonology, Kepler University Hospital, 4021 Linz, Austria; helmut.salzer@kepleruniklinikum.at; 10Division of Mycology, University of Çukurova, 01330 Adana, Turkey; 11Institute of Hygiene and Medical Microbiology, Medical University of Innsbruck, 6020 Innsbruck, Austria; cornelia.lass-floerl@i-med.ac.at

**Keywords:** antifungal resistance, azole, echinocandin, antifungal susceptibility testing, whole-genome sequencing, therapeutic drug monitoring, identification

## Abstract

The high clinical mortality and economic burden posed by invasive fungal infections (IFIs), along with significant agricultural crop loss caused by various fungal species, has resulted in the widespread use of antifungal agents. Selective drug pressure, fungal attributes, and host- and drug-related factors have counteracted the efficacy of the limited systemic antifungal drugs and changed the epidemiological landscape of IFIs. Species belonging to *Candida*, *Aspergillus*, *Cryptococcus*, and *Pneumocystis* are among the fungal pathogens showing notable rates of antifungal resistance. Drug-resistant fungi from the environment are increasingly identified in clinical settings. Furthermore, we have a limited understanding of drug class-specific resistance mechanisms in emerging *Candida* species. The establishment of antifungal stewardship programs in both clinical and agricultural fields and the inclusion of species identification, antifungal susceptibility testing, and therapeutic drug monitoring practices in the clinic can minimize the emergence of drug-resistant fungi. New antifungal drugs featuring promising therapeutic profiles have great promise to treat drug-resistant fungi in the clinical setting. Mitigating antifungal tolerance, a prelude to the emergence of resistance, also requires the development of effective and fungal-specific adjuvants to be used in combination with systemic antifungals.

## 1. Introduction

Fungi are among the most impactful eukaryotic microorganisms living on earth and are associated with enormous crop loss and also to the extinction of various life forms [1]. Apart from being ubiquitously found in the environment, numerous fungal species are considered as part of the normal flora found in different anatomical sites including skin, lung, genitourinary, oral and gastrointestinal tract, where they play an important role in human health [2]. When the immune system is impaired, commensal fungal species can turn into invasive pathogens, translocating systematically to develop invasive fungal infections (IFIs), which affect multiple organs and organ systems [3]. Although underestimated as a cause of infection in humans, fungi are associated with approximately 1.5 million deaths and 1.7 billion superficial infections yearly, resulting in an enormous economic burden [4]. The impacts of fungi on human health go beyond acute and chronic infections, and new lines of study have linked some fungal species colonization to pancreatic cancer progression [5] and alcoholic cirrhosis [6], and we anticipate that future studies using new technologies will unravel unimaginable involvement of fungi in human health.

Fungal species belonging to *Candida*, *Aspergillus*, *Cryptococcus*, and *Pneumocystis* genera are the most clinically relevant pathogens causing IFIs [4]. Unlike the numerous classes of antibiotics used to treat bacterial infections, antifungals are limited in number and belong to three main classes, including azoles (fluconazole, itraconazole, voriconazole, posaconazole, etc.), echinocandins (caspofungin, micafungin, and anidulafungin), and polyenes, such as amphotericin B (AMB) [7]. Azoles bind to *Erg11* in *Candida* and *Cyp51A* in *Aspergillus* species and interrupt the production of ergosterol, a critical sterol component of the fungal cell membrane, while echinocandins target the catalytic subunit of β-1,3-D-glucan synthase, encoded by *FKS* genes and interfere with β-1,3-D-glucan production, a major cell wall structural component [7]. Lastly, polyenes bind to ergosterol in the cell membrane and cause cell death through the formation of large pores on the cell membrane, which leads to interruption of osmotic pressure [7]. Antifungals can be either fungicidal, where the antifungal agent causes fungal cell death, or fungistatic, where the antifungal drug arrests cell proliferation but does not kill the fungal cell, such as fluconazole against *Candida* and echinocandins against *Aspergillus* species. Currently, mold-active triazoles, including voriconazole, posaconazole, and isavuconazole, and echinocandins, especially micafungin and caspofungin, are recommended first-line antifungals used to treat invasive *Aspergillus* and *Candida* infections, respectively [8,9]. Polyenes are used with caution due to the potential for nephrotoxicity and hepatotoxicity and are used more commonly to treat refractory *Candida* and *Aspergillus* infections. Echinocandins are insensitive against *Cryptococcus neoformans* [10]. Therefore cryptococcal infection treatment often involves amphotericin B, 5-fluorocytosine, and fluconazole [11]. Azoles are not the drug of choice for the treatment of *Pneumocystis* pneumonia (PCP), and alternative treatments include dapsone plus trimethoprim, clindamycin plus primaquine, atovaquone, pentamidine, or caspofungin [12]. Moreover, chemoprophylaxis and treatment with trimethoprim–sulfamethoxazole (TMP–SMX) have been associated with a significant reduction of mortality rate among HIV-infected and non-HIV-infected immunocompromised host suffering from PCP [12,13]. Considering the involvement of host- and drug-related factors affecting the antifungal therapeutic failure [14], the emergence of resistance to one antifungal agent can be devastating and severely limit the number of antifungals available to treat IFIs [15].

Prior to the extensive use of antifungals in the clinic (and fungicidal azoles in the environment), epidemiological studies showed the predominance of fungal species almost entirely susceptible to all classes of antifungals. Widespread use of antifungals has altered the epidemiological landscape of IFIs, where fungal species showing resistance to one and/or multiple classes of antifungals are increasingly identified in clinical settings and associated with therapeutic failure [16,17,18,19]. The most notable examples are the worldwide emergence of triazole-resistant *Aspergillus fumigatus* [16,19,20], *Candida tropicalis*, *Candida parapsilosis* [18], and multidrug-resistant (MDR) *Candida auris* [21] and the increasing prevalence of MDR *Candida glabrata*, especially in the U.S. [22]. Beyond *Candida* and *Aspergillus*, a growing body of evidence for *Pneumocystis* indicates that resistance develops in patients receiving sulfa prophylaxis and trimethoprim–sulfamethoxazole [23].

Finally, azole therapeutic failure has been recorded among patients with cryptococcal meningitis during the course of azole therapy [24,25]. Antifungal therapeutic failure can be due to host-, drug-, and fungal-related attributes [26]. For instance, various host underlying conditions, such as abdominal and liver abscesses, prevent the favorable penetration and distribution of antifungal drugs to the site of infection and the resultant suboptimal exposure to antifungal agents promotes survival and emergence of antifungal-resistant fungi [27,28]. Similarly, fungistatic drugs allow fungi to survive in the presence of an antifungal agent, which provides a window for fungi to acquire resistance [7]. Lastly, fungal attributes, such as, but not limited to, acquisition of resistance mutations in the drug target and biofilm formation, allow the fungi to thrive regardless of being exposed to antifungal drugs [7,15]. Of note, resistance to antifungals can be either acquired, where the fungi become resistant during antifungal therapy, or intrinsic, such as *Candida krusei*, which is intrinsically resistant to azoles like fluconazole.

In this review, we summarize the burden of antifungal resistance in the major fungal species associated with IFIs, define the cellular mechanisms underlying antifungal resistance, discuss the phenotypic methods used to distinguish susceptible and resistant fungal isolates, and explore how the application of new technologies, such as whole-genome sequencing, can impact the current paradigms of clinical practice and further of our understanding of antifungal resistance mechanisms. Finally, we address the clinical importance of therapeutic drug monitoring and new antifungal agents in late-stage clinical development and their role to potentially overcome clinical drug resistance. Of note, biofilms, which play an important role in drug resistance, have been extensively reviewed [29,30,31,32,33] and will not be discussed here.

## 2. Epidemiology and Burden of Antifungal Resistance in Clinically Important Fungi

The epidemiology of candidemia and invasive candidiasis, as well as aspergillosis, has been the subject of numerous studies over the years looking to define the scope of fungal burden and antifungal resistance on a global scale. In this context, different studies have investigated the epidemiological trends in *Candida* species infections in which only five species account for 92% of cases of candidemia–*Candida albicans*, *C*. *glabrata*, *C*. *tropicalis*, *C*. *parapsilosis*, and *C*. *krusei*. However, their distribution varies in population-based studies performed depending on the geographical areas (Table 1 and Figure 1) [34]. In a recent work, more than 15,000 invasive *Candida* isolates collected from 39 different countries throughout 20 years were included in a prospective antifungal susceptibility study to analyze the rate of echinocandin and azole resistance. As in the majority of the studies, *C*. *glabrata* was the most commonly detected non-*C*. *albicans* (NAC) species, except in Latin American and some Asian countries, where the NAC predominant species are *C*. *parapsilosis* and *C*. *tropicalis* [18]. The most recent ARTEMIS study revealed an increasing trend with respect to the prevalence of *C*. *glabrata* and *C*. *parapsilosis*. The third and fourth species more commonly isolated, *C*. *tropicalis* and *C*. *krusei*, maintained more steady numbers during the time period studied. Finally, the frequency of other *Candida* species showed an increase of almost 6% [18]. Interestingly, despite being identified as a new agent of candidiasis within the decade, MDR *C. auris* has become increasingly identified as a cause of candidiasis in numerous countries. Shockingly, countries [35] such as India [36] and South Africa [37] have shown a dramatic increase in the prevalence of candidemia due to this MDR *Candida* species. Of particular concern is the tendency of this species to cause clonal outbreaks in clinical settings [38,39,40,41], which likely reflects its resilience on the skin and hospital surfaces, possibly due to biofilm production [42]. These findings support the results obtained in other reports in terms of species prevalence and antifungal resistance [43,44,45,46]. High rates of fluconazole-resistant *C*. *glabrata* (2.8–6.8%) have been reported, together with an increase in *C*. *parapsilosis* (0.6–4.6%), and *C*. *tropicalis* (1.1–9.2%) fluconazole-resistant isolates, apart from the intrinsically fluconazole-resistant species, such as *C*. *krusei* and *C*. *auris* [18]. It is noteworthy that although the worldwide azole resistance rate seems low, such data are extremely worrisome at the institutional level, and not all countries are represented in the ARTEMIS study. Examples are the high level of fluconazole-resistant *C*. *parapsilosis* isolates in Turkey [47] and South Africa [48], *C*. *tropicalis* in Taiwan [49], *C*. *auris* in South Africa [37], and *A*. *fumigatus* in the Netherlands [50].

The in vitro susceptibility of *Cryptococcus* can vary according to serotype, geographical origin and population being studied. Increasingly there are reports of *Cryptococcus* isolates with high azole MIC values, although the correlation of high MICs on clinical outcome is uncertain [51,52]. Clinical breakpoints have not been established for *Cryptococcus* species, although epidemiological cut-off (ECOFF) values have been proposed for some antifungal drugs to differentiate wild-type (WT) from non-wild-type (non-WT) isolates [51], which may result in clinical failure [53].

A systematic review of fluconazole resistance, including a total of 4995 *Cryptococcus* isolates from 3210 patients, showed a mean fluconazole resistance of 10.6% (95% CI: 5.5–15.6) [54]. Relapse isolates showed higher rates of resistance by up to 24%. The vast majority of studies included in the systematic review (28 of 29 studies) defined an ECOFF value of at least ≥16 μg/mL as fluconazole-resistant. MICs above the ECOFF value were reported in 936 of 4995 (18.7%) isolates. Rates of resistance to 5-fluorocytosine among clinical isolates are lower compared to fluconazole [55,56].

Since *P. jirovecii* does not grow in vitro, studies investigating the prevalence of resistance from respiratory samples in HIV-infected and non-HIV-infected immunocompromised patients have mostly employed sequencing and mutation screening in genes involved in drug resistance, including *MT85*, *SOD*, *DHFR* and *DHPS* [57]. The most frequent ones are DHFR312, DHPS165 and DHPS171, but the prevalence of DHPS and DHFR mutations varies significantly, ranging from 0% up to 50% depending on HIV-status and geographical region [58,59]. Most sequence data are coming from single-centers with a limited number of *P. jirovecii* samples. However, the vast majority of data indicate a low prevalence of DHPS and DHFR mutations between 0% and 6% [60,61].

Acquired echinocandin resistance was found to have increased as well [34,103,104]. In the most updated report, the overall resistance rate to one or more echinocandins was lower for most of the *Candida* species, except for *C*. *glabrata* and *C*. *tropicalis* [18]. In contrast to the increasing resistance trend for azoles and echinocandins, several reports showed a broad activity for amphotericin B and no indication of acquired resistance in the species aforementioned [46,105].

Regarding antifungal resistance in *Aspergillus* species, the global emergence of triazole resistance among both clinical and environmental isolates has been increasingly encountered worldwide in the past two decades. In this context, *A*. *fumigatus* is the main etiologic agent of aspergillosis [100,101,106]. The resistance rate for azole-resistant *A*. *fumigatus* varies dramatically by both geographic region and patient cohort and is more prevalent among certain high-risk patient populations [93,107,108], ranging from less than 2% in most of the cases to 30% in some studies which just include hematology and ICU patients [92,93,94,95]. Finally, other human pathogens species of the *Aspergillus* section *Fumigati* also show azole resistance, such as *A*. *lentulus*, *A*. *viridinutans*, *A*. *fumigatiaffinis* and *Neosartorya pseudofischeri,* but due to its prevalence, they do not represent as a notorious global threat as *A*. *fumigatus* infections [103,109]. Therefore, the epidemiological changes and the increased incidence of fungal species exhibiting intrinsic and acquired resistance emphasize the importance of continued surveillance; thus, the proper management of these infections will ultimately lead to improving patient quality of life and survival [18,46].

## 3. Mechanisms of Antifungal Resistance

### 3.1. Antifungal Tolerance

Before discussing the mechanisms employed by fungi to counteract the inhibitory functions of antifungal agents, it is important to distinguish the concepts of antifungal tolerance and resistance.

Antifungal resistance refers to stable genetic changes of a fungal pathogen for a specific class of antifungal drug that results in an increased probability for therapeutic failure. Standardized in vitro antifungal susceptibility testing (AFST) provides a surrogate measure for this potential with drug and organism-specific resistance breakpoints, which take into account mechanism of action, pharmacodynamic responses in animal model systems and clinical response [7]. Antifungal tolerance typically refers to non-heritable cellular adaptations that enable a subpopulation of cells exposed to drugs to persist. Tolerance involves a complex circuitry of signal transduction pathways leading to a rapid and coordinated response to an exogenic agent threatening the cellular integrity and refers to fungal growth higher than susceptible minimum inhibitory concentration (MIC) range in standardized testing. Unlike antifungal resistance, tolerance is a reversible phenomenon meaning that antifungal tolerant cells return to the susceptible MIC range when re-cultured [7]. Pathways involved in tolerance vary depending on the antifungal agents used. For instance, high osmolarity glycerol (HOG), protein kinase C (PKC), and calcineurin are involved in tolerance against echinocandins, which is referred to as cell wall integrity pathway, while PKC and calcineurin are involved in tolerance against azoles, which is defined as membrane integrity pathway [7]. Of particular importance in antifungal tolerance is the heat-shock protein 90 (HSP90), which plays an important role by stabilizing key components of the tolerance, i.e., PKC and calcineurin [110,111]. Therefore, utilization of fungal-specific HSP90 inhibitors in combination with antifungal drugs may have a role as an alternative strategy to effectively combat against the drug-resistant isolates in both molds and *Candida* [110,111,112]. Similarly, combination calmodulin inhibition (fluphenazine) and caspofungin against *C. glabrata* strains carrying F659del in hotspot1 of Fks2p decreased caspofungin MICs, thermotolerance, and the biofilm formation of the strains tested [113]. Similarly, a recent study identified a new protein kinase inhibitor, 2,3-aruyl-pyroazolopyridine, which can abrogate echinocandin resistance of *C. albicans* strains when used in combination with caspofungin [114]. The combination of an echinocandin with an inhibitor of the main components of stress regulatory points to an important route to help overcome and prevent emergence. Although once regarded as an azole AFST artifact, “trailing edges” provide insights into drug tolerance. Newer studies have highlighted the importance of antifungal tolerance and have linked tolerance to both antifungal therapeutic failure and mortality [115,116]. It is now considered a window to drug resistance emergence [117].

When the fungus is exposed to antifungals, tolerance pathways allow cells to persist. The cell’s DNA repair system leads to the occurrence of genetic mutations throughout the genome, especially the drug-target, which is under high selection pressure permits the establishment of stable drug-resistant fungal isolates [22]. Recently, it was observed that almost 35% of *C*. *tropicalis* blood isolates showed a high-level of fluconazole tolerance (≥ 50% of drug-free control), among which almost one-third of infected patients showed therapeutic failure when treated with fluconazole [118]. Moreover, significant variation in tolerance observed among clinical isolates has been associated with a difference in genetic backgrounds [119], which may reflect the mutagenesis potential of a given isolate to induce resistance [22]. Indeed, it was found that some microsatellite clusters contained a significantly higher number of high fluconazole-tolerant *C*. *tropicalis* isolates and vice versa [118]. Unfortunately, despite that there is extensive knowledge about antibiotic tolerance in clinically important bacterial species, our current understanding of antifungal tolerance is still in its infancy. These studies emphasize the clinical significance of antifungal tolerance, and it is recommended to consider the level of tolerance observed (≥50–75% of drug-free control) when choosing an antifungal agent in the clinic [116,118]. It is noteworthy that the existing inhibitory compounds used against stress pathways employed by human fungal pathogens are already used in the clinic and have a profound impact on the host by impairing immune functions [112,120].

### 3.2. Azole Resistance

#### 3.2.1. Azole Resistance in the Candida Genus

Antifungal resistance mechanisms vary depending on the species and the antifungal agents. Generally, changes in drug affinity at the drug target for azoles and echinocandins and overexpression of the drug target and efflux pumps are the major determinants of azole resistance. Considering that azoles are the main drug used to treat invasive candidiasis in developing countries [63,91,121,122,123], having a clear understanding of resistance mechanisms is of paramount importance (Figure 2). Although modification of the drug target, *ERG11*, is an important azole resistance mechanism in species belonging to CTG (*C*. *albicans*, *C*. *parapsilosis*, *C*. *tropicalis*) and Metschnikowia clades (*C*. *auris*) [47,124,125,126], it does not play a prominent role in *C*. *glabrata* [121,127], which is phylogenetically located within the whole-genome duplication (WGD) clade. Heterologous expression and comparative sequencing studies have identified Y132F, K143R, G458S (similar to G464S) as the most prevalent amino acid substitution profoundly impacting the azole susceptibility profiles in CTG and Metschnikowia clades [47,124,125,126]. Of note, there also seems to exist a link between specific mutations in *ERG3* and azole resistance in *C*. *parapsilosis* [128].

Overexpression of drug target and efflux pumps, on the other hand, appears to be a conserved response to azole drugs among species within the three aforementioned clades. Overexpression of *ERG11*, which is controlled by a zinc cluster transcription factor (*UPC2*), results in a higher functional copy number of the drug target and a higher concentration of ergosterol to keep up with the overwhelming concentration of azole drugs [124]. The occurrence of gain-of-function (GOF) mutations, especially in the C-terminal domain of Upc2, results in its structural changes in the ligand-binding domain and its constitutive hyperactivity, followed by nuclear translocation, where it binds to the binding motifs upstream of *ERG11*, subsequently leading to *ERG11* overexpression and overproduction of ergosterol [124,129]. Although its role remains to be elucidated in *C*. *auris*, *C*. *tropicalis*, and *C*. *parapsilosis*, Upc2 can bind to the promoter of *PDR1* and *CDR1* in *C*. *glabrata* [130]. Furthermore, a recent study unveiled a new transcription factor, *Cg*Rpn4, which through the overexpression of genes involved in ergosterol biosynthesis pathways, especially *ERG11*, plays an important role in membrane homeostasis, ergosterol biosynthesis, and azole resistance in *C*. *glabrata* [131]. Chromatin immunoprecipitation assay revealed that *Cg*Rpn4 directly binds to the TTGCAAA binding motif located upstream of *ERG11* [131].

Overexpression of efflux pumps is a common response when fungi are stressed with azoles [124]. Major efflux pumps implicated in azole resistance belong to two major categories, ATP-binding cassette (ABC) transporters (*CDR1* and *CDR2*) and major facilitator superfamily (MFS) transporters (*MDR1* and *MDR2*) [132], mainly regulated by transcription factors Tac1 and Mrr1, respectively [124]. Despite being common in *C*. *albicans* [124], the association between GOF mutations in *TAC1* and *MRR1* and the resultant overexpression of *CDR1* and *MDR1* are poorly studied in *C*. *parapsilosis*, *C*. *tropicalis*, and *C*. *krusei*. Therefore, comprehensive cataloging of *UPC2*, *TAC1*, and *MRR1* in these *Candida* species, followed by their assessment in azole resistance, seems to be a missing knowledge gap in these *Candida* species [133]. Recently, a number of studies cataloged mutations occurring in *UPC2*, *TAC1*, and *MRR1* in azole-resistant *C*. *tropicalis* and *C*. *parapsilosis* isolates [47,118,122,134,135,136], which could serve as a basis to broaden our understanding of azole resistance mechanisms in *C*. *parapsilosis* and *C*. *tropicalis*. New lines of studies identified a prominent role of GOF mutations in TAC1B for azole resistance in *C*. *auris* [111,137]. As for *C*. *glabrata*, several studies have identified specific GOF mutations throughout the *PDR1* gene in clinical isolates of *C*. *glabrata* resistant to azoles [27,127,138,139]. Importantly, the occurrence of such mutations is not only associated with an overexpression of efflux pumps and azole resistance but also a higher virulence due to immunoevasion of *C*. *glabrata* during infection [140,141]. Interestingly, it has been shown that some genes implicated in adhesion, *EPA3*, also play roles in azole resistance by indirectly reducing the intracellular concentration of azole drugs in serially collected clinical isolates of *C*. *glabrata* [139]. Collectively, these studies suggest that mechanisms of azole resistance are complicated and also poorly addressed in some *Candida* species. In light of the increasing prevalence of azole-resistant *C*. *parapsilosis* and *C*. *tropicalis* in numerous countries [18], addressing these mechanisms will have significant implications in drug discovery and the efficacious management of infected patients.

#### 3.2.2. Azole Resistance in *A. fumigatus*

Azole resistance in *A*. *fumigatus* is also a multifactorial phenomenon involving the modification and overexpression of the azole drug target, Cyp51A, and overexpression of efflux pumps, mainly Cdr1B and AtrF [142]. Most prominently, azole resistance results from the occurrence of 34 and 46 bps tandem repeats (TR_34_ and TR_46_) upstream of Cyp51A, which appear to serve as extra binding sites for sterol regulatory element-binding element protein (SrbA) and ABC transporter-regulating transcription factor A (AtrRA) [142]. TR duplications are thought to have an environmental origin and are typically accompanied by mutations in Cyp51A open reading frame, such as TR_34_/L98H and TR_46_/Y121F/T289A, which are also the most prevalent mutations found in azole-resistant clinical and environmental *A*. *fumigatus* isolates [142]. Recent studies have suggested that azole resistance implicated by SrbA and AtrR occurs via overexpression of Cyp51A and Cyp51A and Cdr1B, respectively [143,144]. CCAAT-binding elements, including HapB, HapC, and HapE, bind downstream of the AtrR and SrbA binding sites and negatively regulate Cyp51A expression. Strains containing HapE^P88L^ were shown to be associated with azole resistance in *A*. *fumigatus* clinical isolates lacking Cyp51A mutations [145,146]. A recent study showed that a laboratory-generated voriconazole-resistant *Aspergillus flavus* isolate carried an amino acid substitution in Yap1^L558T^, which was associated with an overexpression of AtrF and voriconazole resistance [147]. New studies implicating various cellular components and pathways involved in azole resistance resulted in significant advances on the topic of azole resistance in *A*. *fumigatus* [148,149,150,151].

### 3.3. Echinocandin Resistance

Unlike complicated resistance mechanisms for azoles involving numerous components, echinocandin resistance is dominated by a single mechanism of action involving mutations in hotspots (HS) regions of *FKS* genes encoding amino acid substitutions within the catalytic subunits of β-1,3-D-glucan synthase [152,153]. As discussed above, echinocandin resistance is relatively rare (<1–2%) in *C*. *albicans* clinical isolates [18,124]. However, more significant echinocandin resistance (2–10%) occurs in *C*. *glabrata*, *C*. *tropicalis*, and *C*. *auris* [18,21]. Epidemiological studies have shown that S629P/T and S663P/F/A are the most observed amino acid substitutions in HS1 of Fks1 and Fks2, respectively, and they are linked to high echinocandin MIC values in clinical isolates of *C*. *glabrata* [133,154]. Among *C*. *albicans* isolates, S456P/F and Ser641P/F are the most prevalent resistance-associated mutations, although other HS1 and HS2 mutations do occur [155]. S645P and S639P/F/Y seem to be the most predominant amino acid substitutions observed in HS1 of Fks1 in echinocandin-resistant clinical isolates of *C*. *tropicalis* and *C*. *auris*, respectively [133]. Clinical isolates of *C*. *parapsilosis* intrinsically have higher MIC values against echinocandins when compared to the other *Candida* species, which is due to a polymorphism P660A in the HS1 of Fks1 [156]. Since bona fide echinocandin-resistant *C*. *parapsilosis* isolates are rarely reported in clinical isolates [18], the respective resistance mechanism remains obscure. Interestingly, a recent report identified four micafungin-resistant *C*. *parapsilosis* blood isolates in Turkey, which harbored a novel amino acid substitution R658G in HS1-Fks1. These isolates were genetically related and resulted in caspofungin therapeutic failure in a patient infected with one of those isolates [157].

The fungistatic action of echinocandins on *Aspergillus* has deterred its broader use on invasive aspergillosis or chronic pulmonary aspergillosis, except for salvage therapy, often in combination with mold-active triazoles [158]. Therefore, echinocandin resistance is quite rare in clinical isolates of *A*. *fumigatus*, but the increasing prevalence of triazole-resistant *A*. *fumigatus* isolates have prompted enhanced use of echinocandin therapy with an expectation for the emergence of resistant isolates. As such, it was documented that F675S in HS1-Fks1 of a clinical *A*. *fumigatus* isolate was acquired following micafungin therapy, which subsequently resulted in therapeutic failure [153]. Interestingly, a new study identified echinocandin-resistant *A*. *fumigatus* isolates that did not harbor any mutations in the *FKS* gene. Authors found a reduction in sensitivity of β-1,3-D-glucan synthase to echinocandins, which was due to prominent lipid changes in the enzyme microenvironment mainly by dihydrosphingosine and phytosphingosine [159]. It is noteworthy that in some cases, mutations just outside of the HS regions can cause echinocandin resistance [160] and also occasionally, echinocandin-susceptible isolates harbor a weak mutation in HS regions that results in therapeutic failure [27]. Therefore, a combination of both *FKS* sequencing and AFST provides the most accurate results.

### 3.4. Antifungal Resistance in Cryptococcus and Pneumocystis

Azole resistance in *Cryptococcus* has been associated with *ERG11* mutations and/ or overexpression of *ERG11* and an ATP-binding cassette (ABC) transporter (*AFR1*). Combination of an efflux blocker (FK506, calcineurin inhibitor) with voriconazole showed four to eight times lower MICs than compared to voriconazole mono in *C*. *neoformans* multi-azole-resistant strains [161]. In multi-azole-resistant strains, the triazole ravuconazole (named BMS-207147 and ER-30346) showed efficacy against strains that encode a protein with a G344S substitution in ERG11 [162]. In *C*. *neoformans*, a small (~1%) heteroresistant subpopulation exists, which, when exposed to drugs, leads to selection and clonal expansion of resistant variants leading to fluconazole resistance [163]. Heteroresistant subpopulations are often not captured in standardized (CLSI or EUCAST) susceptibility procedures. Duplications of chromosome 1 (disomy) are often observed in this resistant subpopulation [164], although *Cryptococcus* undergoes dynamic ploidy changes in response to drug exposure leading to the selection of resistant variants. Resistance occurs due to increases in expression of resistance determinants on Chr1, including *ERG11* and *AFR1*, encoding a major drug efflux transporter. Resistance emergence is related to drug exposure and occurs with the use of clinically relevant regimens but may be potentially overcome by dosage escalation or the use of combination therapy [165].

5-fluorocytosine resistance, on the other hand, is associated with mutations in the UXS1, FUR1 and FCY2 gens and alterations in capsule biosynthesis in *C. neoformans* [166,167]. The intrinsic echinocandin resistance of *Cryptococcus* is not *FKS-*related [168]. While the mechanism of resistance is not fully elucidated, it appears to involve the concerted involvement of *CRM1* and *CDC50* by maintaining cellular calcium homeostasis influencing a mechanosensitive lipid flippase [10], as well as melanin within the capsule, which may act by the complexing drug [169].

As explained above, azoles are not effective in PCP, which was primarily thought to be due to the uptake of sterol from the host lung [170]. However, new lines of studies showed that the ergosterol biosynthetic pathway is active, and ergosterol is also produced by *P. jirovecii* itself. Moreover, the presence of mutations in *ERG11* has been linked to azole resistance in this species [171]. Furthermore, mutations inside two fungus-related genes, namely dihydrofolate reductase (DHFR) and dihydropteroate synthase (DHPS), result in enzyme active site structural changes, which are predicted to interfere with the substrate binding capacity, consequently leading to sulfa resistance [172,173]. Resistance evolves during therapy and can be acquired by person-to-person transmission [11,174]. DHPS mutation can affect clinical outcomes, including a longer duration of mechanical ventilation and increased mortality compared to patients with a wild-type genotype [59,175,176]. Moreover, higher frequencies of treatment-limiting adverse reactions were reported in patients with DHPS [59,175,176]. Therefore, DHPS and DHFR mutational analyses could represent a promising opportunity for optimal patient management, avoiding treatment failure, and finally, death due to PCP. However, more evidence is needed on the frequency of DHPS and DHFR in different patient cohorts and its real impact on clinical outcomes. In a recent pediatric study, phylogenetic analysis revealed 13 unique sequence types (ST), including STs in DHFR and DHPS, that were associated with treatment failures among PCP-positive patients [177]. Yet, this is not universal as a study evaluating a German patient cohort reported a low frequency of such mutations [60].

## 4. Antifungal Susceptibility Testing: Current Paradigm, Challenges, and Solutions

The need for antifungal susceptibility testing (AFST) reflects the increased number of patients having risk factors for invasive fungal infection, the widespread use of antifungals, the rise of acquired resistance, and finally, the emergence of new fungal pathogens [178]. The major goal of AFST is to provide MIC values to guide optimal antifungal therapy and to monitor the emergence and epidemiology of antifungal drug resistance, locally or internationally. However, the drug of choice may be empirically assumed by the proper identification of the fungus and its inherent drug susceptibility profile; hence AFST is recommended when the underlying infection is invasive, and antifungal drug resistance is suspected, or when therapy is failing [179]. The implementation of antifungal stewardship programs is highly recommended, and the current standards in patient care comprise species identification and the performance AFST in isolates collected from sterile body sites, such as blood [180].

Standardized reference methods, the broth microdilution according to the Clinical and Laboratory Standards Institute (CLSI) and the European Committee on Antimicrobial Susceptibility Testing (EUCAST) represent the gold standards for AFST, but they are time-consuming and labor-intensive [179,181,182,183,184]. Alternative tests or commercial products include disk diffusion, epsilometer tests, colorimetric broth microdilution, and automated spectrophotometric systems [178]. Recently, a new low-cost four-well plate agar method is suitable for echinocandin susceptibility screening of *Aspergillus* species and can be used to detect echinocandin non-WT isolates [185]. Moreover, MIC test strip and Sensititre YeastOne showed an overall agreement of >72% and >94% at ±1 and ±2 dilutions when compared to CLSI and EUCAST broth microdilution methods, respectively [186].

All these AFST methods display a variety of specific advantages and disadvantages. In general, AFST is time-consuming as providing MICs usually takes 24–48 h from the time point culture is available. In addition, the interpretation of results may be difficult for fungi where clinical breakpoints are lacking. The lack of interpretive criteria for various fungus-antifungal combinations raises the question of how to deal with such data and implement them in clinical practice [181]. Besides these practical issues, it should be noted that technical (aerobic environment, high glucose, planktonic cells, growth media), as well as fungal factors (a mixture of hyphae, conidia and fungal biofilms, fungal virulence and fitness) may influence MIC readings [187]. Of note, *P. jirovecii* does not grow on culture media in vitro, which precludes the utility of AFST for PCP.

New approaches to aid AFST target culture-independent techniques, such as MALDI-TOF MS technology and molecular-based resistance detection. The call for molecular-based susceptibility testing is growing, which aims to detect validated genetic resistance mechanisms. Such analyses may provide results within a few hours, but various limitations (detailed in Section 5) may complicate interpretation [178]. Some MALDI-TOF MS assisted AFST methods have been evaluated for a limited number of fungal species and drugs [188,189]. The MALDI-TOF MS is promising but needs a simplified, automated method for testing a broad range of fungi and antifungals.

## 5. Emerging Molecular Approaches to Diagnose Resistance, Current State and Challenges

The ability of a fungal cell to stably resist drug exposure and transfer it to its progeny is encoded in its genome [190]. This fact implies that, if the genetic determinants of resistance are known, a genetic test can be developed to detect the presence of such a trait. This is the norm for assessing drug susceptibility in viruses and many bacteria. Similarly, for medical mycology, this is a very promising approach given the significant investment of time necessary to perform standard drug susceptibility testing (discussed in Section 4). In addition, the availability of molecular approaches to amplify or probe specific DNA sequences, as well the continuous developments in next-generation sequencing (NGS), are paving the way for more specific, accurate, and cost-effective approaches [191]. Despite much progress, these approaches have achieved limited implementation in clinical mycology [191,192,193].

A major limitation of molecular approaches for the diagnosis of resistance is that the same resistance phenotype can be caused by mutations at different loci [27,123,140]. Thus, previous knowledge of the resistance-causing mutations is a prerequisite for developing an accurate technique. Once a set of resistance-conferring mutations is known, specific tests for their presence can be developed based on PCR amplification of nucleic acid hybridization approaches, or whole-genome sequencing data can be compared to a database of known mutations [191]. All these approaches have been successfully used for different *Candida* and *Aspergillus* pathogens [194,195,196,197,198]. Along the same line, commercial real-time PCR assays have been developed for the identification of DHPS mutations [199]. As discussed in Section 3, this knowledge is highly limited for fungal pathogens such as *C*. *parapsilosis* and *C*. *tropicalis* [133], but, as research progresses, this gap is expected to be closed, at least for the most common pathogens. Probe-based approaches have higher specificity and are thus promising for direct resistance screening of clinical specimens. However, they require a catalog of known resistance-conferring mutations, which is only partially available even for the best-studied species. The more incomplete that catalog of resistance-conferring mutations is, the more false-negatives will this test produce. Direct sequencing approaches allow the discovery of novel mutations–although they still require independent validation linking a genetic change to the resistance phenotype for application. Whole-genome sequencing can provide very accurate results in organisms for which the resistance-conferring mechanisms are well understood. For instance, a genomic survey of bacterial isolates recovered from clinical samples has shown an excellent categorical agreement with standard antimicrobial susceptibility testing [200,201]. Using this standard, echinocandin resistance among *Candida* species is ideally suited for resistance assessment, as is azole resistance among *Aspergillus*, as there is a strong correlation between clinical resistance, MIC and specific mutations in target genes [133]. Once a comparable level of knowledge is achieved for the different fungal pathogens and drugs, it is envisaged that a similar level of accuracy could be obtained.

Most applications explored so far have been focused on target sequencing of specific regions or whole-genome sequencing of isolates. However, emerging NGS applications include the direct sequencing of entire communities from clinical samples using a whole-genome shotgun approach coupled to a computer-based reconstruction of complete or partial genome sequences of the organisms present in the sample–, i.e., metagenomics [202,203]. For instance, independent of resistance, a recent study successfully employed a high-resolution metagenomics approach to a fecal sample using internal transcribed spacer 1 to predict candidemia [3]. Although more expensive and involving steeper requirements for downstream analyses, a metagenomics approach has several advantages. First, it is entirely unbiased and does not require a prior hypothesis on the infecting organism, whose identity and resistance potential can be diagnosed from the analysis of the sequencing reads. Second, as it is not based on a single isolate, it can inform on the genetic diversity of the infecting population at the sampled sites. Lastly, in addition to the information on the infecting agent, it can provide information on the host and the entire microbial community present in the sample site, which can also have clinical relevance [204]. Increasing throughput and cost-effectiveness of sequencing machines and the emergence of long-read sequencing technologies, such as oxford nanopore, are making metagenomics approaches more affordable and informative. One could envision a metagenomic approach being used in a prognostic manner, surveying stool, blood, or saliva samples from patients in order to monitor the presence of drug-resistant microbes and thereby anticipate treatment failure and allowing or preparing for therapy adjustments before a systemic infection with a resistant strain appears.

All these promising developments notwithstanding, the implementation of NGS technologies in a clinical setting is still very far from being routine. There are many challenges that need to be overcome, including cost, time, equipment and expertise [159]. Although sequencing costs and even sequencing equipment is decreasing, the bulk of the cost for NGS applications is related to both sample preparation (DNA extraction, library preparation) or to bioinformatics analysis of the data. Both procedures require either highly trained personnel or costly automated alternatives. In addition, the amount of time elapsed from sample collection to specific information needs to be minimal (<24 h) to facilitate clinical value. This (and possibly some ethical considerations) limits outsourcing analyses to most external providers. Finally, although NGS provides a wide array of interesting information that could be exploited in the future, in the clinic, this information needs to be highly focused and processed to provide direct answers to specific clinical questions so that clinicians can quickly react to the data instead of being overwhelmed. Ideally, a clinical-grade NGS system should be encapsulated in the form of a fully or semi-automated device that processes a sample and outputs actionable data in the most direct and rapid way. Such devices, like the Cepheid Xpert MTB/RIF, have revolutionized the management of *Mycobacterium tuberculosis* (MTB) infections by providing faster and more accurate MTB diagnosis that detects MTB and rifampicin (RIF) resistance simultaneously [205]. The push for personalized medicine in other clinical fields such as cancer or rare diseases is driving the penetration of genomics in the clinic, and it is likely that current limitations, at least in terms of costs and resources, will eventually be overcome. In sum, as our knowledge of the genetic basis of antifungal resistance grows and matures, and as new molecular tools enter a mainstream microbial diagnosis, it is anticipated that the molecular detection of antifungal resistance will be an important component of clinical care.

## 6. Therapeutic Drug Monitoring and Drug Dose Optimization

The treatment of invasive fungal infections is often complex and complicated by multiple factors that may affect the absorption, distribution, and metabolism of antifungal agents, including predisposing factors for infection and the site and severity of the infection. Therapeutic drug monitoring (TDM) is utilized to measure antifungal drug levels to prevent inadequate dosing that may lead to the emergence of resistance and/or treatment failure or excessive dosing that may lead to drug toxicity. The major guidelines recommend TDM once a steady-state has been reached for triazole-based treatment for invasive aspergillosis (IA) and when extended courses of prophylaxis are used, including by the Infectious Diseases Society of America [9], European Society for Clinical Microbiology and Infectious Diseases (ESCMID) and the European Confederation of Medical Mycology (ECMM) [206], and the European Conference on Infections in Leukemia (ECIL) [207]. The European Respiratory Society (ERS) and ESCMID also recommend TDM for triazole-based treatment for chronic pulmonary aspergillosis [208]. The IDSA also recommends TDM with voriconazole during the treatment of invasive candidiasis (IC) [8], as does the ESCMID when 5-fluorocytosine, voriconazole, and posaconazole are used [209].

In vitro studies and animal modeling have shown that the area under the concentration-time curve and minimum inhibitor concentration (AUC/MIC) ratio is the pharmacokinetic/pharmacodynamics (PK/PD) index that is most predictive of posaconazole efficacy against yeasts [210]. In invasive mold infections, the time above the MIC (T>MIC) and the AUC/MIC ratio are particularly important with posaconazole antifungal therapy [211] and other triazoles [212], as well as for echinocandins. In an analysis of 6 of 10 clinical trials of patients receiving voriconazole, there was no association between voriconazole levels and treatment efficacy, though, because antifungal exposure far exceeded the MICs of most pathogens [213]. The peak concentration (C_max_) and MIC are particularly important PK/PD indices for amphotericin B and the echinocandins [214]. Lastly, the T>MIC is the most relevant PK/PD indices for flucytosine [214].

While sub-therapeutic drug levels increase the risk for treatment failure, they also can increase the risk for resistance emergence with invasive fungal infections for patients on antifungal prophylaxis or extended treatment durations. For example, multiple studies have shown that sub-therapeutic trough concentrations during posaconazole prophylaxis in patients with hematologic malignancies are associated with breakthrough invasive fungal infections [215,216,217]. Studies of lung transplant recipients on prophylactic voriconazole have found higher rates of fungal colonization and breakthrough fungal infections [218] in those with sub-therapeutic voriconazole drug levels. In addition, higher rates of breakthrough fungal infections have been observed in patients with sub-therapeutic voriconazole levels following allogeneic stem cell transplantation in patients [219].

As stated above, acquired azole-resistance in *A*. *fumigatus* is becoming increasingly problematic [220,221,222,223], partly driven by the use of triazoles as environmental fungicides [224,225]. In addition to sub-therapeutic drug levels, increasing the risk of breakthrough invasive fungal infections, there is also a risk of causing increased antifungal resistance. In one study of patients with cystic fibrosis of an azole therapy, primarily being used for the treatment of allergic bronchopulmonary aspergillosis (ABPA), persistently positive fungal culture, or *Aspergillus* bronchitis, azole levels were sub-therapeutic in half of the patients, with an association between sub-therapeutic drug levels and increased antimicrobial resistance [226]. Even in the treatment of azole-resistant infections, using higher doses of voriconazole [227,228] and posaconazole [229] has been shown to be effective in the treatment of these infections. Thus, the effective use of TDM may help prevent the emergence of drug-resistant fungal infections and facilitate the treatment of azole-resistant infections, possibly with higher doses of a given antifungal, although more investigation of this strategy is needed.

## 7. Optimizing Therapy by Species Identification

As discussed, fungal species respond differently to various antifungal agents; therefore, species identification plays a pivotal role in drug-decision-making practices, which is endorsed by standardized clinical treatment practices [8]. The most notable example is the stark azole susceptibility pattern observed between the highly susceptible *C*. *albicans* and inherently resistant *C*. *krusei.* Unfortunately, accurate and rapid identification tools, such as matrix-assisted laser desorption ionization time-of-flight and Sanger sequencing, which are common in Western countries, are largely unaffordable for most clinical laboratories of developing countries, and most of the identifications are performed by inaccurate and expensive phenotypic tools [123]. To help overcome this issue, a comprehensive multiplex PCR was introduced [230], which can identify the most clinically important yeast species causing infection in humans and numerous studies afterward showed its reproducibility in various epidemiological studies [92,231,232]. Although such techniques are far from ideal, their application can significantly shorten the turnaround time and improve the species identification accuracy. We hope that the future advances in technology and materials used in MALDI-TOF MS and Sanger sequencing offer affordable options to be used in developing countries.

## 8. Therapeutic Challenges and how to Overcome Them

New antifungal drugs have been scarce over the last decade, and only one new antifungal (a member of the class of triazoles) has been approved for clinical use. The urgency for new antifungal classes is growing as cases have increased, and as detailed above, once-treatable fungi are becoming resistant. Furthermore, only one approved class of antifungal drugs, the azoles, can be taken orally. Due to increasing resistance rates against azoles and echinocandins, invasive *Candida* infections have become more difficult to treat, given the limited number of classes of antifungals currently available. This limitation in antifungal treatment options was prominently highlighted by the emergence of *C*. *auris*, a multidrug-resistant species, which has been associated with outbreaks worldwide and led to clinical alerts to U.S. and European healthcare facilities [233,234]. In the U.S., 90% of *C*. *auris* isolates appear resistant to fluconazole, about 30% have been resistant to amphotericin B, and less than 5% have been resistant to echinocandins, while about one-quarter of recent Indian isolates of *C*. *auris* were resistant to two or more classes of antifungals [233,234,235]. For the treatment of these highly resistant *Candida* species, approval of ibrexafungerp, a member of a new antifungal class, which, like the echinocandins, attacks the fungal cell wall, but does so by latching onto another part of glucan synthase, is eagerly awaited. Ibrexafungerp is currently evaluated in a clinical phase III trial [236] and shows excellent oral bioavailability, making the drug potentially very valuable for the longer-term or ambulant treatment of yeast infections resistant to azoles or in patients with contraindications against azoles. Rezafungin is a novel echinocandin, for which phase III clinical trials are about to start, which has a much longer half-life than other echinocandins, allowing it to be given via a once a week injection instead of daily [237,238]. Most importantly, it can be delivered at high dosages, which overcome certain types of echinocandin resistance. Rezafungin has improved activity against *Aspergillus* species compared to the other echinocandins and may, therefore, become an alternative option for treating azole-resistant aspergillosis [237].

The emergence of triazole resistance complicates the selection of appropriate antifungal treatment for invasive aspergillosis, where triazoles are considered the first-line treatment option [239,240,241,242,243]. Given that more areas may be burdened with high rates of environmental triazole resistance, triazoles may not be universally recommended as primary antifungal treatment, but instead, treatment choice may depend on the local epidemiology of azole-resistant *A*. *fumigatus*. While triazole resistance is considered an emerging threat for patients infected by *A*. *fumigatus* [244], leaving very limited treatment options for those patients, triazole resistance in *Aspergillus terreus* [245] may be even more threatening because *A*. *terreus* can be non-susceptible to amphotericin B. Liposomal amphotericin B is the primary alternative option to azoles for treatment of IPA, and therefore the drug of choice for treatment of azole-resistant aspergillosis, but as stated above nephrotoxicity may occur [9]. Alternative options for second-line treatment of azole-resistant aspergillosis are echinocandins, which show, however, reduced activity against *Aspergillus* when used as monotherapy [236]. In regions where environmental triazole resistance rates of *Aspergillus* exceed 10%, primary treatment of all invasive aspergillosis cases with either liposomal amphotericin B or echinocandin-voriconazole combination has been recommended, with a later step down to voriconazole should the isolate show voriconazole susceptibility [183]. New antifungal classes currently under clinical development, including fosmanogepix, also known as APX001 is a first-in-class and orally available broad-spectrum antifungal agent, which targets the highly conserved Gwt1 fungal enzyme, and olorofim, which belongs to a new class of antifungals called the orotomides and targets dihydroorotate dehydrogenase in the de novo pyrimidine biosynthesis pathway) [246], have high potency against *A*. *fumigatus* without the same burden of drug–drug interactions and toxicity [247], and may therefore overcome the limitations of currently available antifungals for azole-resistant aspergillosis and become the preferred treatment options in the near future. Moreover, to accelerate drug development and accelerate regulatory approvals (e.g., Food and Drug Administration (FDA)), investigators are looking to repurpose existing FDA-approved compounds developed against other diseases with the hope of identifying efficacious new antifungal agents that can be rapidly deployed [248,249]. Interestingly, some of these compounds not only proved to be efficacious against drug-susceptible fungal species [248,249] but also some showed desired efficacy when tested on MDR *Candida* and mold species [250,251]. Therefore, the future battle against the drug-resistant fungal species in the clinic may benefit from comprehensive repurposing of FDA-approved compounds.

## 9. Future Directions

The increasing frequency of drug resistance in *Candida* and *Aspergillus* poses a serious threat to human health as there remains a limited number of systemic antifungal drugs available to treat IFIs. The increasing prevalence of azole resistance among invasive NAC species, especially *C*. *parapsilosis* and *C*. *tropicalis,* highlights a pressing need to better understand the underlying resistance mechanisms involved. Minimizing the global emergence of azole-resistant *A*. *fumigatus* isolates requires serious measures regarding antifungal stewardship in both agriculture and the clinic. Accordingly, the application of AFST, TDM, and accurate species identification is recommended as a part of routine antifungal drug decision-making. Finally, limited antifungal drug classes and the increasing presence of drug-resistant fungi should direct our efforts to continuously identify new drug classes with promising in vivo activities, preferably acting on cellular mechanisms apart from current systemic antifungals. New agents with novel mechanisms of action in late-stage clinical development will be welcome to help address this need. Furthermore, adjuvant drugs targeting HSP90-, PKC-, and calcineurin-inhibitors have the potential to limit drug tolerance to slow resistance emergence, and their development is encouraged.

## Figures and Tables

**Figure 1 antibiotics-09-00877-f001:**
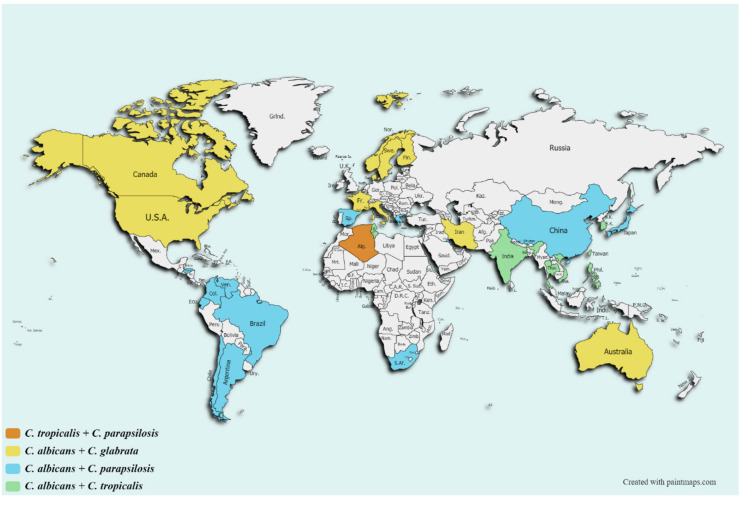
Worldwide prevalence of non-*albicans Candida* species causing candidemia.

**Figure 2 antibiotics-09-00877-f002:**
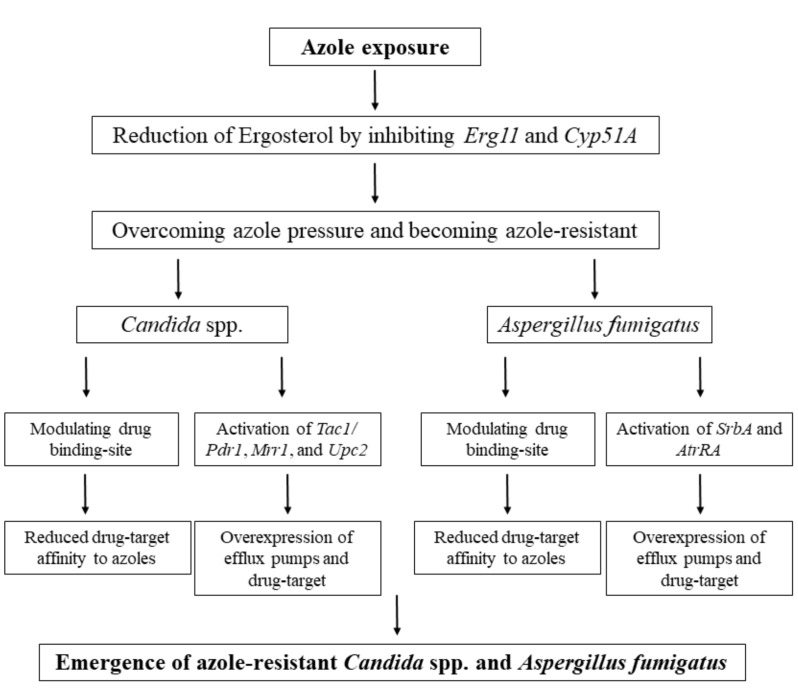
General mechanisms of azole resistance mechanisms employed by *Candida* and *Aspergillus* species covered in this study. The drug-target modulating and overexpression of efflux pumps and drug target are the most common strategies counteracting the azole efficacy used by these species.

**Table 1 antibiotics-09-00877-t001:** The epidemiology of candidemia and invasive aspergillosis and antifungal resistance rates determined for selected *Candida* species and *Aspergillus fumigatus*.

Species	Prevalence (%)	Azole Resistance Rate (%)	Echinocandin Resistance Rate (%)	References
***C*. *albicans***	>20.9–70	0–7.8	0–7	[43,44,46,48,62,63,64,65,66,67,68,69,70,71,72,73,74,75,76,77,78,79,80,81,82,83,84,85,86,87]
***C*. *glabrata***	<15	0–21	0–23.1	[48,62,67,69,70,74,75,76,78,84]
	15–38	0–76	0–100	[43,44,46,63,66,68,71,73,77,79,80,81,82,83,85,86,87]
***C*. *tropicalis***	<10	0.6–31.5	0–8	[62,63,64,65,66,67,69,70,71,82,83]
	10–49	0–66.6	0–10	[43,44,46,48,68,72,73,74,75,76,77,78,79,80,81,84,85,86,87]
***C. parapsilosis***	<15	0–8.3	0–10.8	[46,62,63,77,78,79,80,81,82,85]
	15–37	0–53	0–1.6	[43,44,48,65,71,73,74,75,76,83,84,86,87]
***C. auris***	0–14	>90–100	0–40	[38,62,88,89,90,91]
***A. fumigatus***	33.2–92	<2–30	0	[92,93,94,95,96,97,98,99,100,101,102]

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
