# Peer review of "Drug-Resistant Fungi: An Emerging Challenge Threatening Our Limited Antifungal Armamentarium"

_antibiotics, 2020, doi:10.3390/antibiotics9120877_

Round 1

Reviewer 1 Report

The review covers the topic of antifungal drug resistance in human pathogenic fungi in the genera Aspergillus and Candida.  This is a concern globally, particularly with the lack of new antifungal agents that could be used on these resistant strains.  One limitation of the article is the scope of coverage.  Although the authors (lines 103-104) suggest that they “summarize the burden of antifungal resistance on the major fungal species associated with IFIs…” they report only on two genera.  They do acknowledge that recent reviews have been written about the Cryptococcus species, however, at least some summary of those findings and how that might be integrated into the review from section 5 onwards would be appropriate given the title of the article.  What about resistance in Histoplasma capsulatum or Pneumocystis?   Candida auris, however interesting as a newly emerged species, can not be considered a “major” species causing IFI. The authors would be perhaps better to consider the narrative of the review more carefully by either expanding the species described or perhaps tackling a compare-and-contrast between what is basically A. fumigatus and the Candida species.

The manuscript would benefit with an additional round of editing.  A few points are:

Line 59: go for goes.

Lines 185-188: cells plural.  The sentence on DNA repair system is unclear, and sounds like Lamarckian evolution.

Line 207: delete comma after Although.

Line 209: which is phylogenetically.

Line 231: are for is.

Lines 238-239: “missing knowledge gap” yet “numerous studies”.

Line 246: delete as a consequence.

Lines 260-261: “to be acquired from the environment” is unclear.

Line 288: delete comma.

Line 300: fumigatus that did not.

Line 301-304: unclear, recommend rewriting.

Line 319: sterile body site requires clarification, as there should not be microbes there.

Line 336: within a few hours.

Author Response

The review covers the topic of antifungal drug resistance in human pathogenic fungi in the genera Aspergillus and Candida.  This is a concern globally, particularly with the lack of new antifungal agents that could be used on these resistant strains.  One limitation of the article is the scope of coverage.  Although the authors (lines 103-104) suggest that they “summarize the burden of antifungal resistance on the major fungal species associated with IFIs…” they report only on two genera.  They do acknowledge that recent reviews have been written about the Cryptococcus species, however, at least some summary of those findings and how that might be integrated into the review from section 5 onwards would be appropriate given the title of the article.  What about resistance in Histoplasma capsulatum or Pneumocystis?   Candida auris, however interesting as a newly emerged species, can not be considered a “major” species causing IFI. The authors would be perhaps better to consider the narrative of the review more carefully by either expanding the species described or perhaps tackling a compare-and-contrast between what is basically A. fumigatus and the Candida species.

Response: Thanks for raising such thoughtful comments. We do agree that the number of fungal species with intrinsic and/or acquired resistance is expanding. Species within Histoplasma and Pneumocystis, and also other dimorphic fungi, although clinically-relevant, they are more endemic and restricted to specific areas in the world and indeed require dedicated studies where authors can highlight various aspects like virulence, resistance, epidemiology, etc. Therefore, herein, we aimed at discussing resistance mechanisms and relevant actions in fungal species with a global expansion and not endemic to specific areas. Regarding C. auris, although primarily was known as a rare yeast species, it not only has a global spread as one of the most important multidrug-resistant Candida species with worldwide distribution, but also now it is one of the leading cause of candidemia in certain countries, such as India (doi: 10.1111/myc.12790.) and South Africa (doi: 10.3201/eid2411.180368. and doi: 10.3201/eid2509.190040.), and replacing highly prevalent Candida species. The importance of this species is further evidenced by its resilience to stay on the surfaces and its tendency to cause clonal outbreaks.

The manuscript would benefit with an additional round of editing.  A few points are:

Line 59: go for goes.

Lines 185-188: cells plural.  The sentence on DNA repair system is unclear, and sounds like Lamarckian Response: Thanks for correcting the typos throughout the text. We appreciate it. Correction was applied.

evolution.

Response: Correction was applied. The sentence was modified to clearly convey what authors intended. Indeed, it seems like Lamarckian evolution theory, however, here with a single cell scale.

Line 207: delete comma after Although.

Response: correction was applied.

Line 209: which is phylogenetically.

Response: correction was applied.

Line 231: are for is.

Response: correction was applied.

Lines 238-239: “missing knowledge gap” yet “numerous studies”.

Response: Thanks. Indeed, there was a contradiction. Correction was applied.

Line 246: delete as a consequence.

Response: correction was applied.

Lines 260-261: “to be acquired from the environment” is unclear.

Response: Clarifications were applied.

Line 288: delete comma.

Response: Correction was applied.

Line 300: fumigatus that did not.

Response: Correction was applied.

Line 301-304: unclear, recommend rewriting.

Response: This sentence was changed to clearly convey what authors intended.

Line 319: sterile body site requires clarification, as there should not be microbes there.

Response: Example was included.

Line 336: within a few hours.

Response: Correction was applied.

Reviewer 2 Report

The manuscript discussed about the invasive fungal infections and the information is interesting for the readers. However, major issue of in the present manuscript is poor way of presentation.

  • Author should add a detailed table that contained the literature of clinically important Candida and Aspergillus spp.
  • Author should add a world map of IFI, to provide a reader friendly introduction.
  • Author should add at least 2-3 pictorial diagram or schematic diagram of antifungal resistance, Emerging molecular approaches to diagnose resistance, current state and challenges and how the infection spread and causing impact of crop yield.

Author Response

Reviewer 2:

The manuscript discussed about the invasive fungal infections and the information is interesting for the readers. However, major issue of in the present manuscript is poor way of presentation.

Author should add a detailed table that contained the literature of clinically important Candida and Aspergillus spp.

Author should add a world map of IFI, to provide a reader friendly introduction.

Author should add at least 2-3 pictorial diagram or schematic diagram of antifungal resistance, Emerging molecular approaches to diagnose resistance, current state and challenges and how the infection spread and causing impact of crop yield.

Response to all three queries raised: Thanks indeed, we believe that addition of figures and tables are quite important and would significantly enrich the context of the current review paper. Therefore, we have added one figure showing the worldwide distribution of Candida species. Unfortunately, we could not add Aspergillus on this figure, since most of the countries already had Candida and colored in accordance with prevalence of Candida species. To compensate this, we included a comprehensive table, which included both prevalence and resistance rate of Candida and Aspergillus species. We also included a diagram showing the paradigm of azole resistance in both Candida and Aspergillus species. Of note, we did not add echinocandin resistance on this figure, since the mechanism is straightforward and is caused by occurrence of mutation in FKS gene. Regarding the emerging diagnostic approaches, we already included one section dedicated to this topic (section 5). The impact of fungi on crop loss, however, was not in line with the scope intended for the current review, which has been recently reviewed somewhere else in details (The world’s ten most feared fungi).

Reviewer 3 Report

A poorly detailed manuscript. Some parts in my opinion make you lose context appropriateness. Chapters 4 and 5 are absolutely out of context. Chapters 6 and 7 are undersized, as well as 8.

It is not clear what the authors mean in chapter 4. Many of the statements contained are absolutely spurious, cannot be criticized without then suggesting the remedy in depth and discussing its advantages.

What did the authors want to write a review or an author opinion? If they were thinking of a review, the entire manuscript must be rewritten and discussed in more detail. Above all by deepening both the Cilinic context of invasive forms of fungi and a greater discussion on the actors of these infections.
There is no careful reflection on the pharmacokinetics and pharmacodynamics of antifungals and the role of any associations. Why no mention of the off-label use of some antifungals precisely in invasive forms?
If this is an author opinion then what conclusions are absolutely incomplete

Specific comments
line 54 "where they play an important role in human health" the sentence does not match the content of the bibliographic citation

Line 96 to 97 "Similarly, fungistatic drugs allow the fungi to survive
in presence of antifungal agent and provide a window for fungi to acquire resistance "this sentence in relation with bibliographic citation is not clear. Do the authors want to refer to tolerance?

Author Response

Reviewer 3:

A poorly detailed manuscript. Some parts in my opinion make you lose context appropriateness. Chapters 4 and 5 are absolutely out of context. Chapters 6 and 7 are undersized, as well as 8.

Response: We appreciate for the time dedicated by reviewer to evaluate our review. As reviewer also stated below, we worked on sections 6 and 7 and added contents offered by reviewer to improve the limitations. We are afraid that we hold an entirely different viewpoints for sections 4 and 5 and authors believe that addition of these sections will present a more comprehensive perspective on the topic of antifungal resistance. To address specifically, section 4 is dealing with the current phenotypic methods used to identify antifungal resistance, which is an indispensable topic when discussing antifungal resistance. Moreover, this section has highlighted the drawbacks of currently used phenotypic methods and highlighted some roadblock to be paved in the future. The next section, section 5, is presented in tandem with the shortages of the current phenotypic and PCR-based molecular methods and offers novel platforms, such as whole-genome sequencing and metagenomics, to tackle such shortages as well as describes the technical issues on the way of such methodologies. Both sections have been prepared by features scientists in the field of medical mycology and we believe that presenting such differences in viewpoint will be always helpful in creating discussions as well as establishing the basis for the future works.

It is not clear what the authors mean in chapter 4. Many of the statements contained are absolutely spurious, cannot be criticized without then suggesting the remedy in depth and discussing its advantages.

Response: The comment was dealt with above and the section was reviewed by the co-author responsible for this section.

What did the authors want to write a review or an author opinion? If they were thinking of a review, the entire manuscript must be rewritten and discussed in more detail. Above all by deepening both the Cilinic context of invasive forms of fungi and a greater discussion on the actors of these infections.
There is no careful reflection on the pharmacokinetics and pharmacodynamics of antifungals and the role of any associations. Why no mention of the off-label use of some antifungals precisely in invasive forms?
If this is an author opinion then what conclusions are absolutely incomplete.

Response: We do respect the viewpoints that reviewer holds. The clinical relevance of the species included are clearly displayed in introduction and section 2, which also has been reviewed elsewhere extensively (doi.org/10.1371/journal.pntd.0007964, doi: 10.1128/mBio.00449-20, doi: 10.15698/mic2020.06.718). We do admit that the PK/ PD is lacking and the relevant section was added to section 6. We also expanded section seven by including some comments with respect to off-label antifungals, which also has been most recently reviewed by others (doi: 10.1039/c9cs00556k.).

Specific comments

line 54 "where they play an important role in human health" the sentence does not match the content of the bibliographic citation

Response: The influence of commensal fungi on human health is presented in the very beginning of the abstract as well as the introduction “Intestinal domination by Candida species has been shown to be a major source of Candida bloodstream infections. Fungal dysbiosis is also linked to the development and treatment response in non-fungal infections, for example Clostridioides difficile colitis and HIV.”.

Line 96 to 97 "Similarly, fungistatic drugs allow the fungi to survive in presence of antifungal agent and provide a window for fungi to acquire resistance "this sentence in relation with bibliographic citation is not clear. Do the authors want to refer to tolerance?

Response: Yes, we meant tolerance. But it also should be noted that tolerance can also happen in the presence of fungicidal drugs, for instance the echinocandin tolerance exerted by Candida glabrata (DOI: 10.1101/2020.08.14.251181).

Reviewer 4 Report

This is a well considered and structured review on the topical subject area of anti-fungal drug resistance. It is quite scientifically orientated and goes into a lot of detail, for example on mechanisms of antifungal drug resistance.

Yet, it also includes useful information on antifungal susceptibility testing and on therapeutic drug monitoring  both of which will be of interest to clinicians 

The way it is written though is, at times, confusing and doesn't facilitate the fluency needed for a seamless read.

For example on page 2 there is a statement: "Similar to Aspergillus species echinocandins are shown to be insensitive against Cryptococcus neoformans"

The above is grammatically incorrect and potentially confusing.

On the top of page 3 there is a discussion about treatment for pneumocystis infection. A list of drugs is provided yet the agent most commonly used, co-trimoxazole, is not mentioned until a subsequent statement.

Later on page 3 there is a discussion on drug target mutations in DHPS and DHFR. Are the authors referring to human host mutations and not in fungi?

A further example of a confusing sentence is on page 7: "Unfortunately, unlike the wealth knowledge exists regarding the antibiotic tolerance...."

There are quite a few other examples of confusing sentences or misspellings

It might be worth considering an additional Table detailing the main resistance mechanisms for the leading fungal pathogens as discussed in section 3 which would facilitate quick reference

Author Response

Reviewer 4

This is a well considered and structured review on the topical subject area of anti-fungal

drug resistance. It is quite scientifically orientated and goes into a lot of detail, for example

on mechanisms of antifungal drug resistance.

Yet, it also includes useful information on antifungal susceptibility testing and on

therapeutic drug monitoring both of which will be of interest to clinicians

The way it is written though is, at times, confusing and doesn't facilitate the fluency needed

for a seamless read.

For example on page 2 there is a statement: "Similar to Aspergillus species echinocandins

are shown to be insensitive against Cryptococcus neoformans"

The above is grammatically incorrect and potentially confusing.

Response: Thanks for your note, this has already been revised.

On the top of page 3 there is a discussion about treatment for pneumocystis infection. A list

of drugs is provided yet the agent most commonly used, co-trimoxazole, is not mentioned

until a subsequent statement.

Response: Thanks for your note, this has already been revised.

Later on page 3 there is a discussion on drug target mutations in DHPS and DHFR. Are the

authors referring to human host mutations and not in fungi?

Response: It refers to the fungus and relevant changes have been applied accordingly.

A further example of a confusing sentence is on page 7: "Unfortunately, unlike the wealth

knowledge exists regarding the antibiotic tolerance...."

Response: This sentence was corrected per your suggestion.

There are quite a few other examples of confusing sentences or misspellings

It might be worth considering an additional Table detailing the main resistance mechanisms

for the leading fungal pathogens as discussed in section 3 which would facilitate quick

reference

Response: We do agree and that is why the same concept has been already included in Figure 2.

Round 2

Reviewer 1 Report

My original main comment ("The authors would be perhaps better to consider the narrative of the review more carefully by either expanding the species described or perhaps tackling a compare-and-contrast between what is basically A. fumigatus and the Candida species.") has really not been addressed, so this is a major issue still with the manuscript.  The authors have suggested they are only looking at species that are not "restricted to specific areas of the world", although do not include this as part of the describing the scope of the manuscript.  Furthermore, Pneumocystis, Cryptococcus, Histoplasma and probably others are of global distribution, so should be included. 

Author Response

Response: Thanks for asking for further clarifications. We extensively revised our paper per reviewer’s comment and added respective information for both Cryptococcus and Pneumocystis throughout the text. In addition, in order to respond to referee’s queries several papers have been included in the text as well we explored the references and inserted with them updated ones.

Reviewer 4 Report

The authors have addressed earlier concerns. The manuscript would benefit overall from careful editing to rectify persisting grammatical errors which somewhat spoil the read

Round 3

Reviewer 1 Report

The major concern about the contents of the manuscript continue, in that this is not a review about all fungi, but really only on Aspergillus and Candida species.  Certainly there is now some additional text on Cryptococcus and Pneumocystis, but that is not integrated into the rest of the text.

Author Response

Response: We do appreciate the concern of the reviewer and the current version included an updated information on Cryptococcus and Pneumocystis. An extensive English editing has also been provided.